# Sybil Attacks Detection and Traceability Mechanism Based on Beacon Packets in Connected Automobile Vehicles

**DOI:** 10.3390/s24072153

**Published:** 2024-03-27

**Authors:** Yaling Zhu, Jia Zeng, Fangchen Weng, Dan Han, Yiyu Yang, Xiaoqi Li, Yuqing Zhang

**Affiliations:** 1The School of Cyberspace Security, Hainan University, Haikou 570208, China; zhuyl@nipc.org.cn (Y.Z.); zengj@nipc.org.cn (J.Z.); hand@nipc.org.cn (D.H.); yangyy@nipc.org.cn (Y.Y.); csxqli@gmail.com (X.L.); 2The National Computer Intrusion Protection Center, University of Chinese Academy of Sciences, Beijing 101408, China

**Keywords:** CAVs, Sybil attacks, traceability, attacker, security

## Abstract

Connected Automobile Vehicles (CAVs) enable cooperative driving and traffic management by sharing traffic information between them and other vehicles and infrastructures. However, malicious vehicles create Sybil vehicles by forging multiple identities and sharing false location information with CAVs, misleading their decisions and behaviors. The existing work on defending against Sybil attacks has almost exclusively focused on detecting Sybil vehicles, ignoring the traceability of malicious vehicles. As a result, they cannot fundamentally alleviate Sybil attacks. In this work, we focus on tracking the attack source of malicious vehicles by using a novel detection mechanism that relies on vehicle broadcast beacon packets. Firstly, the roadside units (RSUs) randomly instruct vehicles to perform customized key broadcasting and listening within communication range. This allows the vehicle to prove its physical presence by broadcasting. Then, RSU analyzes the beacon packets listened to by the vehicle and constructs a neighbor graph between the vehicles based on the customized particular fields in the beacon packets. Finally, the vehicle’s credibility is determined by calculating the edge success probability of vehicles in the neighbor graph, ultimately achieving the detection of Sybil vehicles and tracing malicious vehicles. The experimental results demonstrate that our scheme achieves the real-time detection and tracking of Sybil vehicles, with precision and recall rates of 98.53% and 95.93%, respectively, solving the challenge of existing detection schemes failing to combat Sybil attacks from the root.

## 1. Introduction

Traffic congestion and accidents are common problems faced in metropolitan areas. In the United States, according to the statistics of the National Highway Traffic Safety Administration (NHTSA) [1], billions of traffic waiting times cause the unnecessary consumption of more than 3.1 billion gallons of fuel each year. On the other hand, about 35,000 people are killed, and nearly 4 million people are injured due to traffic accidents, and the average annual economic loss is more than USD 836 billion. Therefore, governments and researchers actively seek solutions, such as more intelligent roads and traffic signals. With the expansion and extension of Internet applications and the support of the new generation of information technology represented by 5G, CAV technology can effectively solve the above problems. According to a report by Allied Market Research, the global self-driving car market is estimated to be USD 54.23 billion in 2019 and is expected to reach USD 556.67 billion by 2026 [2].

However, as CAVs grow, the cybersecurity risks they face are becoming more pronounced. The Upstream 2022 report shows that over 900 CAV cybersecurity incidents occurred in 2021 alone [3]. This grim reality has drawn the close attention of many researchers and prompted them to explore in depth the security threats faced by CAVs to propose effective prevention and response strategies [4,5,6,7,8,9,10,11]. Among the many threats, Sybil attacks [12] are particularly challenging. During information sharing and cooperative driving by CAVs, attackers forge multiple identities or location information and launch this relatively low-cost attack by broadcasting false data. In this attack mode, a malicious vehicle can control or influence a large number of normal nodes by using only a small number of nodes, posing a serious hazard to the CAV system. Therefore, it is considered one of the top threats in Telematics. In order to deal with the potential threat of Sybil attacks in CAV networks, we must conduct more in-depth research on attack detection and countermeasure strategies. Benadla et al. have provided a brief description of the impact of Sybil attacks on vehicular networks and a detailed categorization of Sybil attack detection methods in VANETs [13]. These studies provide valuable references but still need to be further explored in depth in order to establish a more complete and effective defense mechanism.

During information sharing and cooperative driving between CAVs, an attacker may create and broadcast false data and thus launch a Sybil attack. In such attacks, attackers aim to control or influence a large number of normal nodes using only a small number of nodes by forging multiple identities or location information. In the CAV networks, a malicious vehicle is a physical vehicle that can obtain multiple legitimate identities illegally, and it is fully capable of launching Sybil attacks by forging vehicle location information and simulating the operating characteristics of normal vehicles. As a result, the road condition monitoring and decision-making of the CAV systems will be easily confused and misled, causing DOS attacks [14] on the CAV systems, even leading to traffic accidents, casualties, and property losses. As shown in Figure 1, before the Sybil attack is launched, CAVs know that the current traffic is relatively smooth through information sharing among them. As depicted in Figure 2, malicious vehicles broadcast traffic packets with false information to interfere with the CAV driving status. This leads some vehicles to misinterpret traffic conditions as being more congested, resulting in reduced speed or lane changes, thus causing inefficiency in the entire traffic fleet. Once a normal vehicle trusts a Sybil vehicle, a malicious vehicle can successfully mislead a normal vehicle. Therefore, the detection of Sybil attacks in CAVs is necessary.

Once the normal vehicles trust the Sybil vehicles, malicious vehicles can successfully mislead normal vehicles. Therefore, the detection of Sybil attacks in CAVs is necessary. In addition, the collusion [15] and separation behaviors between malicious and Sybil vehicles make malicious vehicles exhibit similar behavioral characteristics as normal vehicles. This is the main reason why many mitigation solutions against Sybil attacks can only detect Sybil vehicles but cannot accurately track malicious vehicles [16,17,18,19,20,21]. Notably, malicious vehicles are the source of Sybil attacks. Yang et al. organized Sybil attacks regarding the traceability of malicious vehicles [22], pointing out that existing schemes have apparent drawbacks for the traceability of malicious vehicles. This allows malicious vehicles to continue to exist in the vehicular network and continue Sybil attacks. Zhang et al. [23] also suggested that accurately tracking malicious vehicles is one of the pressing challenges in solving Sybil attacks. Currently, most of the schemes rely on RSSI values to accomplish the detection. Yuan proposed an edge computing-based Sybil detection scheme [24], where the vehicle sends a control packet to two nearby edge nodes. The nodes use the Jake model to calculate the RSSI, transmit it to each other, and determine the range of normal RSSI ratios through multiple rounds of computation to detect malicious vehicles. However, RSSI values are susceptible to real-world scenario factors such as traffic, attacker density, etc. Krishnan et al. proposed a collaborative strategy to detect malicious vehicles [25]. Since malicious vehicles need to maintain more Sybil vehicles, they are more likely to have the longest list of nearest vehicles. However, this scheme mainly targets Sybil attacks in the presence of a single malicious vehicle, and its detection rate decreases once there is a conspiracy between malicious vehicles. Rakhi et al. proposed a Sybil detection method based on LCSS similarity computation and RSSI time-series variation point detection [26]. Without power control, malicious vehicles are detected by finding the similarity of RSSI nodes. However, when the distance between vehicles is relatively small, RSSI sequences received from normal and malicious nodes show high similarity, and it is difficult to identify malicious nodes from normal nodes. To address the single detection factor limitation, Chen et al. proposed a multi-scale data fusion detection framework for the Sybil attack [27]. By acquiring BSM, map data, and sensor data, the detection of malicious vehicles is accomplished using machine learning classification models. However, the limitation is that the framework can only be used in the exact location where the Sybil attack occurred. Secondly, the detectors laid on the roads are costly. Finally, since the scheme incorporates machine learning, when the attack samples in the training dataset with low attack density are much smaller than normal samples, it will prevent the model from learning the attack behavior sufficiently, resulting in lower accuracy and recall. While attempting to address the problem of malicious vehicle tracing, these works have limitations, as they mainly rely on RSSI values or machine learning that requires extensive training. In contrast, our research is based on a vehicle broadcast beacon packet detection mechanism that is not susceptible to traffic and attack density and only requires a few training samples. We can only stop Sybil attacks from launching Sybil attacks at their source by accurately identifying malicious vehicles.

We observe that a key feature of Sybil attacks is that Sybil vehicles are essentially beacon data packets forged by malicious vehicles. These fake vehicles lack the broadcasting and listening capabilities of real ones and must rely on malicious vehicles to mimic the behavior of normal ones. Given this, we propose a beacon packet-based detection method to trace malicious vehicles based on trusted RSUs combined with information interactions performed between CAVs. Firstly, the roadside unit (RSU) enables vehicles within the detection range to communicate with each other in a point-to-multipoint (PMP) manner and exchange a specific field in the beacon data packet to prove their physical existence. We refer to this specific field in the beacon packet as *Key*. If the exchange of *Key* is successful, it is considered that both communication parties are physical nodes, and we build an edge for them in the adjacency matrix. Note that since Sybil vehicles do not have actual broadcasting and listening capabilities, this will result in the inability to establish edges between Sybil vehicles and normal vehicles. Therefore, we can detect Sybil vehicles simply by traversing the neighbor relationships in the adjacency matrix. At this point, the RSU will assign the Sybil vehicle a *Key* that identifies its identity, and the Sybil vehicle will share the *Key* with the malicious vehicles. In order to maintain the existence of Sybil vehicles, malicious vehicles will then help broadcast the *Key* on behalf of Sybil vehicles, making normal vehicles believe that Sybil vehicles also have the broadcasting capability, mistakenly. Therefore, we can trace the malicious vehicles based on the *Key* sharing behavior between the Sybil vehicles and the malicious vehicles. The traceability of malicious vehicles can curb the occurrence of Sybil attacks from the source.

After classification using the Sybil vehicle detection algorithm, we perform a fine-grained classification of “original Sybs” and “original Hons”. First, “original Sybs” are categorized at a fine-grained level based on the probability of failure to communicate successfully between Sybil vehicles and normal vehicles. Second, Original Hons are categorized at a fine-grained level based on the key-sharing behavior between Sybil vehicles and malicious vehicles. The fine-grained classification process accurately identifies malicious vehicles while improving the detection rate of Sybil vehicles. Our detection mechanism can detect Sybil vehicles and trace malicious vehicles in real time and efficiently under different Sybil attack densities, malicious vehicle densities, and vehicle density attacks.

The main contributions of this work are as follows.
•We propose a beacon packet-based scheme to trace malicious vehicles. The F1 of the malicious vehicle reaches 96.38%, which helps to resist the Sybil attack from its source.•The detection rate of Sybil vehicles is improved while tracing the malicious vehicles, and the F1 for Sybil vehicles reaches 98.11%, which is 2% higher than in the literature [17].•Neighborhood graphs are formed instantaneously and independently without reference to the vehicles’ historical trust values, reducing the privacy risk associated with historical data.

The full text of this paper is organized as follows: Section 2 introduces the progress of related research work; Section 3 elaborates on our threat models; Section 4 introduces our tracking mechanism for malicious vehicles; Section 5 is our experimental part; Section 6 compares the performance of our proposed method with existing methods; and Section 7 is the summary and outlook of the full text.

## 2. Related Work

Much research has recently been conducted on Sybil attacks at home and abroad. Nonetheless, many schemes have focused only on detecting Sybil vehicles faked by malicious vehicles. Existing detection schemes for detecting Sybil vehicles can be divided into two categories: direct identity detection mechanisms and indirect identity detection mechanisms.

### 2.1. Direct Identity Detection Mechanisms

Direct identity detection mechanisms mainly achieve identity detection through the authentication of vehicle certificates or keys [28]. Santhosh [29] proposed a hybrid cryptographic management mechanism to detect Sybil vehicles. In this scheme, the base station generates a public key for the vehicle that wants to enter the network to communicate and uses the node’s identity to encrypt the public key. Nevertheless, this scheme does not consider the security threat to the private key during the generation process. In order to solve this problem, Cheng [30] proposed an RSU authentication scheme based on elliptic curve cryptography, which effectively reduces the security threats of generating pseudonyms and private keys. Regardless, there are overhead and delay problems when RSU authenticates a large number of vehicle beacon packets. Although the direct identity detection schemes [31,32,33] are the most direct way to verify the identity of illegal vehicles in real time, they cannot meet the needs of rapid authentication when vehicles are gathered in a short time or move at high speeds. In addition, since malicious vehicles have legitimate identities, schemes through direct identity detection fail to detect malicious vehicles.

### 2.2. Indirect Identity Detection Mechanisms

Compared with the overhead of direct identity detection mechanisms, indirect identity detection mechanisms are more lightweight, such as the detection methods based on vehicle cooperation [18,21,34,35,36]. Panchal [37] proposed a method of separating Sybil attacks using adjacent information in VANETs. In this scheme, RSU discovers its neighbor vehicles through the vehicle ID and then evaluates the trust degree of neighbor vehicles to detect Sybil attacks. However, this scheme cannot avoid the situation of avoiding detection due to collusion between Sybil vehicles. It is also impossible to avoid the credibility evaluation of the vehicle itself. If it is a malicious vehicle, the credibility of its neighbor vehicle list is unconvincing. In order to solve that reliability problem of the vehicle itself, Huo [38] designed and implemented an identity authentication mechanism based on vehicle history information. The vehicle generates the historical message hash value *HD1* from the historical message sent by itself and then hashes the message to be sent and the historical information into *HD2* to send to the agent. The agent detects the Sybil vehicle by comparing *HD2* with *HD1*. However, the historical messages in this scheme have too much influence on future decision-making, and it is impossible to avoid the long-term latency of malicious vehicles. To solve the problem of relying on historical messages, Verchok [17] proposed a scheme for detecting Sybil nodes to verify the existence of each other through local peer-to-peer communication. In this scheme, the server randomly instructs any pair of nodes within the communication threshold to perform a customized beacon packet broadcast and listen. The beacon packet contains a *Key* that identifies the vehicle’s identity, and the success or failure edge is established by whether the *Key* is exchanged successfully. Finally, a neighbor graph is formed. Due to the unequal information between Sybil nodes and normal nodes, the formed neighbor graph will make Sybil nodes extreme, and then Sybil nodes can be detected.

The existing solutions mainly focus on detecting Sybil vehicles because malicious vehicles in Sybil attacks typically exhibit normal behavior, which increases the difficulty of detecting malicious vehicles. Angappan [39] proposed a Sybil attack detection scheme that combines RSSI and neighbor information, detects Sybil attacks by comparing the similarity of RSSI, and regards the vehicle with the most neighbor entries as a malicious vehicle. Yet, multiple vehicles may have the same RSSI value only if these vehicles belong to the same vehicle, which cannot defend against collusion in Sybil attacks. In addition, since power affects RSSI, this scheme can only detect malicious vehicles without power control but cannot detect malicious vehicles with transmission power control [40]. Zhang [41] proposed a detection method based on Basic Safety Message (BSM) packets in 2023, which mainly detects the detected vehicle by receiving the receiver (*Recv*) of the BSM message broadcast by the detected vehicle. The *Recv* calculates the transmission distance ($T_{d}$) of the BSM message according to the sending time and receiving time information of the BSM message. Next, the distance ($D_{N_{i}}$) between the surrounding neighbor vehicles ($N_{i}$) and *Recv*, respectively, is calculated. Finally, we take the $N_{i}$ with the minimum error between $D_{N_{i}}$ and $T_{d}$ as *Recv* predicts the BSM package broadcast source. Finally, an integral strategy is implemented for all receivers ($Recv_{i}$) of the BSM message, and the neighbor vehicle $N_{i}$ with the highest integral value is selected as the final predicted broadcast source of the BSM message broadcast by the detected vehicle. Sybil vehicles and malicious vehicles are detected by comparing whether the predicted broadcast source is consistent with the broadcast source marked in the BSM message. Regardless, the defect of this scheme is that the proposed spatio-temporal model is prone to errors when it is affected by vehicle densities.

In summary, existing Sybil attack detection schemes mainly focus on detecting Sybil vehicles. However, only by tracing malicious vehicles can Sybil attacks be resisted from their source. Compared with existing solutions, our proposed traceability mechanism considers collusive behavior and separation behavior between malicious vehicles. It does not need to analyze many vehicle trajectories, nor is it affected by vehicle densities to complete the detection of Sybil vehicles and trace back to malicious vehicles, resisting attacks from the source of Sybil attacks.

## 3. Threat Model and Attack Categories

First, we describe the assumptions on which this study is based. Next, our threat model is presented. Finally, the attack types of Sybil attacks are explored.

### 3.1. Assumptions

Our research is based on the following assumptions:All actual vehicles are equipped with a high-precision Global Positioning System.The RSU has the power to indicate the vehicle’s broadcast or listening status.Keys listened to by malicious vehicles can be shared between malicious vehicles, creating opportunities for other malicious vehicles in a listening state to maintain Sybil vehicles [17].The success rate of communication between the malicious vehicle and the Sybil vehicle is set to 0.6 [17].

### 3.2. Threat Model

In order to attack successfully, malicious vehicles usually use illegal means to obtain legitimate identities. While the literature [20] mentions that malicious attacks by vehicles with illegal identities can be prevented by Public Key Infrastructure (PKI), vehicles with legal identities cannot be prevented from broadcasting false information. Our application scenario is that all vehicles have legitimate identities.

We classify the vehicles into three categories: normal vehicles, malicious vehicles and Sybil vehicles. The properties of the vehicle are shown in Table 1.

We define a vehicle with a physical location as an actual vehicle. In our research, normal vehicles only broadcast authentic and credible beacon packets related to themselves. Although malicious vehicles broadcast authentic and credible beacon packets related to themselves, in order to control the driving status of normal vehicles, they often also broadcast some fake beacon packets in the VANETs. These beacon packets are Sybil vehicles.

In this research, we do not need to consider the authentication of illegal vehicle identity when the vehicle joins the VANETs, and the direct identity detection schemes have already solved this problem. Instead, our detection mechanism focuses on detecting the Sybil attacks launched by the attacker by sending false location messages after entering the VANETs. This kind of internal attack often causes more damage to the VANETs.

The purpose of malicious vehicles launching Sybil attacks is to obtain network resources that are disproportionate to normal vehicles by forging Sybil vehicles. Furthermore, it is to use a small number of vehicles to control and influence as many normal vehicles in the VANETs as possible, and finally decide on the control systems. Figure 3 is an example of an attacker model, where vehicles *N1*, *N2*, and *N3* are normal vehicles, and *M1* is a malicious vehicle.

As shown in Figure 3, *M1* forges a beacon packet at time *t1* and broadcasts it to *N1*, *N2*, and *N3*, where the pseudonym (*S1*) in the beacon packet is a valid pseudonym obtained by *M1* through illegal means. *Pos* is the forged position information of *M1*. When *N2* receives the beacon packet, it thinks that the vehicle *S1* is also in the same lane. In order to avoid collision, *N2* takes a deceleration or emergency braking operation. After analyzing the beacon packets, *N1* and *N3* find that the road conditions they are on are not affected, and they continue to drive normally. Finally, *M1* successfully interferes with the driving state of *N2* by generating a Sybil vehicle (*S1*) by forging its location.

### 3.3. Attack Categories

We consider the collusion and separation behaviors of malicious vehicles as different attack categories.
Collusion behaviors: Malicious vehicles can share information through particular internal communication.
(a)Malicious vehicles help Sybil vehicles broadcast. For example, when a Sybil vehicle is instructed to broadcast by the RSU, in order to reduce the possibility of the Sybil vehicle being exposed, the nearby malicious vehicle in the listening state will use the identity of the Sybil vehicle instead of the Sybil vehicle to perform the *Key* broadcast so that normal vehicles mistakenly believe that the Sybil vehicle also has normal communication ability.(b)Malicious vehicles help Sybil vehicles listen. For example, when the RSU instructs a Sybil vehicle to listen, the nearby malicious vehicles will share the listened lists with the Sybil vehicle, making the RSU mistakenly believe that the Sybil vehicle also can listen.Separation behaviors: We consider the separation behavior between malicious vehicles and Sybil vehicles and set the communication success rate between malicious vehicles and Sybil vehicles to 0.6 [17], which can reduce the clustering between malicious vehicles and Sybil vehicles, which is more in line with the selfish behavior of malicious vehicles in Sybil attacks.

## 4. Beacon Packet-Based Traceability Mechanism

A beacon packet is a type of data packet used in vehicle-to-vehicle communication technology. We have customized the format of the beacon packet message as {*PseudonymID*, *Time*, *Key*, *Pos*}, where *PseudonymID* is the pseudonym ID of the vehicle broadcaster; *Time* is the sending time of the beacon packet; *Key* is a particular field allocated by RSU to the vehicle in the broadcast state, where we believe that the *Key* is only known by the vehicle itself and the RSU and cannot be forged; and *Pos* is the location information when the vehicle broadcaster broadcasts the beacon packet, which can be forged by malicious vehicles.

In this study, we mainly complete the detection of Sybil attacks based on the location of the vehicles and further trace the malicious vehicle on this basis. In the first step, we apply the Sybil detection algorithm proposed in the literature [17] to the CAVs and achieve good results. Through a large number of experiments, we find that the detection algorithm is not sensitive to the detection of malicious vehicles. It means that malicious vehicles can continue to launch Sybil attacks at the right time, and the security risks to VANETs still exist. Our research aims to complete the traceability of malicious vehicles and to resist Sybil attacks from the source to improve vehicle safety and traffic efficiencies.

### 4.1. Execution of Communication

In order to allow each vehicle to have sufficient opportunities to communicate with each other, RSU adopts a dichotomy method to assign the broadcast or listen state to the vehicle (the broadcast state is assigned a *Key* that can identify the vehicle’s identity, and the listen state is assigned “listen”). First, the RSU divides the vehicles into approximately equal broadcast and listen groups. In the first round, half of the vehicles are set to broadcast and the other half to listen. In the second round, the vehicle status is reversed. After every two rounds, RSU divides the subgroups divided in the previous round and performs state distribution so that each vehicle can have a sufficient opportunity to communicate with other vehicles within the time complexity of log (*N*). Among them, *N* is the number of detected vehicles.

We use an example to explain the process of RSU assigning a broadcast or listen status to vehicles using the dichotomy method. As shown in Figure 4, the vehicles to be detected are *A*, *B*, *C*, and *D*. In the first round, the vehicles to be detected are dichotomized. The subgroup composed of *A* and *B* is set as the broadcast state, the assigned particular fields are *K1* and *K2*, respectively, and the subgroup composed of *C* and *D* is set as the listening state. In the second round, the vehicle status is reversed. The subgroup composed of *A* and *B* is in the listening state, the subgroup composed of *C* and *D* is in the broadcasting state, and the assigned particular fields are *K3* and *K4*, respectively. In the third round, the subgroup composed of *A* and *B* is dichotomized again, and subgroup *A* is set to be in the broadcast state. The particular field assigned is *K5*, and the subgroup *B* is in the listening state. In the fourth round, the vehicle’s status is reversed, subgroup *A* is in the listening state, subgroup *B* is in the broadcasting state, and the assigned particular field is *K7*. Similarly, we perform the dichotomy method on the subgroups formed by *C* and *D*.

### 4.2. Construction of the Neighborhood Graph

After the communication execution in Section 4.1 is completed, all the vehicles to be detected report to the RSU the *Key* lists they have listened to, and the RSU builds neighbor edges for vehicles according to the *Key* lists and finally completes the neighbor graph construction. The neighbor graph comprises all the vehicles (*N*) to be detected and the edges *E* formed between the vehicles, that is, G=(N,E), where $N=\left\{N_{1},N_{2}\ldots ,N_{n}\right\}$, *E* is the directed edges formed between two vehicles, and $E_{\left(i,j\right)}$ represents the directed edges from $N_{i}$ to vehicle $N_{j}$. If $N_{j}$ listens to the *key* of $N_{i}$, it means that $N_{i}$ can form a successful edge to $N_{j}$, that is, $E_{\left(i,j\right)}$ = *S*; otherwise, a failed edge is formed, which is recorded as $E_{\left(i,j\right)}$ = *F*.

We use an example to describe the process of RSU constructing a neighborhood graph for a vehicle as shown in Figure 5. Assume that *N1* and *N2* are normal vehicles, *M* is a malicious vehicle, and *S* is a Sybil vehicle generated by *M*. With *S* not maintained by *M*, its interactions with *N1* and *N2* will fail. The failure from *N2* to *M* is due to the communication failure caused by environmental factors. The success from *S* to *M* is the communication success of the malicious vehicle with a probability of 0.6 to reduce the possibility of Sybil vehicle exposure. The failure from *M* to *S* is a separation behavior between the malicious vehicle and the Sybil vehicle, and the malicious vehicle fails to communicate with the Sybil vehicle to reduce the cluster.

### 4.3. Construction of Probabilistic Neighborhood Graph

With Section 4.2, we can now construct a neighbor graph for vehicles. Next, we have to transform the neighbor graph into a probabilistic neighbor graph, which in turn computes the vehicle’s confidence level.

#### 4.3.1. Distances and Probabilities Relationships

The traceability mechanism in this article is based on broadcast communication between vehicles, involving signal attenuation during wireless communication [1]. Newport [42] proposed that the probability of beacon reception does indeed decay with the distance between the transmitters and receivers. To explore the relationship between vehicle distances and communication success, we experiment with two cars, *A* and *B*. Vehicle *A* broadcasts beacon packets at a frequency of 0.05 s in place. In contrast, vehicle *B* moves at 10 m/s in the opposite direction, allowing us to observe how well vehicle *B* received these packets. We consider signal fading caused by paths and obstacles and analyze the packet loss rate every 50 m. As shown in Figure 6, we find the relationship between the vehicle distances and the communication success probabilities.

According to the distances between vehicles, interpolating the distance intervals for the proposed Distances and Communication Success Probabilities model is the probability of success of the edges between vehicles.

#### 4.3.2. Probabilistic Neighborhood Graph

Consider that when a vehicle broadcasts, it forms an outgoing edge for itself, and when it listens, it forms an incoming edge. In each round, the number of outgoing edges formed as broadcasters is much smaller than that of incoming edges formed as listeners. Therefore, we analyze the input edge set with more data than the output edge set to reduce the errors.

According to the Distances and Communication Success Probabilities model mentioned in Section 4.3.1, we consider the probability of edge success as a function of the distances between vehicles. Assuming that the vehicle ($N_{j}$) listens to the probability of the vehicle ($N_{i}$) broadcasting the beacon packet expressed as $P_{\left(i,j\right)}$, the calculation is shown in Equation (Equation 1):(1)$$\begin{array}{ccc} P_{\left(i,j\right)} & = & Prob[dist\left(E_{\left(i,j\right)}\right)] \end{array}$$
where $E_{\left(i,j\right)}$ represents the incoming edge formed from $N_{i}$ to $N_{j}$, dist() is a function to calculate the distance between $N_{i}$ and $N_{j}$, and Prob[] is the probability of successful communication based on the distance between vehicles combined with the Distances and Communication Success Probabilities model in Section 4.3.1. Therefore, the calculation of the success or failure probability of $E_{\left(i,j\right)}$ is shown in Equation (Equation 2):(2)$$\mathrm{Edge}\left(E_{\left(i,j\right)}\right)=\left\{\begin{array}{cc} P_{\left(i,j\right)}, & E_{\left(i,j\right)}=S\\ 1-P_{\left(i,j\right)}, & E_{\left(i,j\right)}=F \end{array}\right.$$

The total probability product of a vehicle’s incoming edge set during the entire communication process is shown in Equation (Equation 3):(3)$$\begin{array}{ccc} P_{Total} & = & \prod_{i\neq j}^{N}Edge\left(E_{\left(i,j\right)}\right) \end{array}$$

For the convenience of calculation, the logarithm of $P_{total}$ is calculated and recorded as $P_{Pval}$ as shown in Equation (Equation 4):(4)$$\begin{array}{c} Pval=\ln\left(P_{Total}\right)=\sum_{i\neq j}^{N}\ln\left(Edge\left(E_{i,j}\right)\right) \end{array}$$

We refer to Pval as the vehicle’s credibility after calculating the credibility of each vehicle by Equation (Equation 4). The neighbor graph in Section 4.2 is transformed into a probabilistic neighbor graph as shown in Figure 7.

It can be seen from Figure 7 that since *S* itself has no broadcast and listen capabilities, the possibility of forming a successful edge with *N1* or *N2* is tiny, which leads to the low credibility of *S*. At the same time, since the communication success rate between *M* and *S* is 0.6, this will also cause the credibility of *M* to be affected by *S*.

### 4.4. Traceability Mechanism of Malicious Vehicles

After we apply the Sybil vehicle detection algorithm in [17] to our traceability mechanism, we can divide the vehicles into two categories: the original predicted Sybil vehicle group (“original Sybs”) and the original predicted normal vehicle group (“original Hons”). However, after many experiments, we find that there are wrongly predicted vehicles (such as malicious vehicles or normal vehicles) in “original Sybs”; similarly, there are also wrongly predicted vehicles (such as malicious vehicles and Sybil vehicles) in “original Hons”. In addition, we also find that the algorithm has a poor detection rate for malicious vehicles, which means that malicious vehicles may relaunch Sybil attacks at the right time. Therefore, it is impossible to eradicate Sybil attacks from the root, which is also our focus: to detect “original Sybs” and “original Hons” further to detect malicious vehicles and improve the precision of Sybil vehicles.

Our traceability mechanism is divided into two stages. The first stage subdivides the “original Sybs”, and the second stage subdivides the “original Hons”. The traceability mechanism for malicious vehicles is shown in Figure 8:

#### 4.4.1. Subcategory “Original Sybs”

Vehicles misclassified in “original Sybs” may be malicious or normal vehicles. Malicious or normal vehicles have normal broadcast and listening capabilities. For Sybil vehicles, without the maintenance of malicious vehicles, they will not have normal broadcast and listening capabilities. Therefore, our strategy is first to screen out vehicles with normal broadcast and listen capability from “original Sybs” as Predictive Normal and Malicious Vehicle Groups (*PreHon&Mal*). The remaining vehicles are classified as Predictive Sybil Vehicle Group 1 (*PreSybs1*). Secondly, *PreHon&Mal* is further subdivided into Predictive Malicious Vehicle Group 1 (*PreMals1*) and Predictive Normal Vehicle Group 1 (*PreHons1*).

Because normal vehicles cannot communicate with Sybil vehicles but they can communicate with malicious vehicles and normal vehicles, we need to select a certain number of normal vehicles to screen out Sybil vehicles. According to the Pval mentioned in Section 4.3.2, the larger the Pval, the more likely the vehicle is a normal vehicle. We use the parameter $\alpha_{\mathrm{Tru}-\mathrm{Nodes}}$ = 0.1 to select vehicles with higher Pval values, record them as *Tru-Nodes*, and collocate them as listening states. Since setting all “original Sybs” to the broadcast state can easily raise suspicion of malicious vehicles, we use the dichotomy method mentioned in Section 4.1 to allocate the communication state of “original Sybs”. Without the malicious vehicles’ help, Sybil vehicles do not have broadcast capability, and *Tru-Nodes* cannot listen to the key broadcast by Sybil vehicles. After sufficient communication between the “original Sybs” and *Tru-Nodes* groups, RSU extracts and analyzes the list of listened-to keys uploaded by *Tru-Nodes* to determine the list of vehicles listened to by each trusted vehicle.

Considering that there is a communication failure probability of 0.4 between malicious vehicles and Sybil vehicles, this may lead to the Pval value of such malicious vehicles not being affected by Sybil vehicles and then appearing in the selected *Tru-Nodes*. When *Tru-Nodes* fully communicate with the “original Sybs”, there is still a communication success probability of 0.6 between malicious vehicles and Sybil vehicles, which leads to the malicious vehicles being able to listen to the majority of Sybil vehicles in the “original Sybs”. In addition, malicious vehicles in the listening state may also help Sybil broadcast, allowing normal vehicles to listen to Sybil vehicles, which may cause this part of normal vehicles to be misclassified as suspiciously malicious. Therefore, we use the parameter $\beta_{similar}$ = 0.9 to ensure that all extracted *Tru-Nodes* are normal vehicles as much as possible. If the number of “original Sybs” listened to by a vehicle in the *Tru-Nodes* is greater than Len(Original Sybs)*$\beta_{similar}$, the vehicle is considered a suspicious mal. The remaining vehicles after these vehicles are eliminated constitute a SuplsToppval and reassign the communication status of the SuplsToppval. Otherwise, *Tru-Nodes* are considered trustworthy vehicles. At this point, the vehicles listened to by *Tru-Nodes* are classified as *PreHon&Mal*, and the remaining vehicles are *PreSybs1*. Further classifications of misclassified vehicles can help improve the accuracy of Sybil vehicles. The process of using *Tru-Nodes* to screen *PreHon&Mal* with broadcast capability is shown in Algorithm 1.
**Algorithm 1** Screening vehicles that can broadcast.detectCanBdcastNodes(nodes,IdToPval)    CanBdcastNodes←[]    *n*←len(IdToPval)∗0.1    For
i=0→n−1       TopPvalId←IdToPval[i]    EndFor    NodeSum←nodes+TopPvalId    $Rounds\leftarrow 2*\log_{2}\left(len\left(NodeSum\right)\right)$    For
i←Rounds       For
j←TopPvalId          If
CommPlan[j,rnd]≠ ”listen”              broadkeys.APPEND(ConnSim[j,rnd])          Else              ListenedKeys.APPEND(ConnSim[j,rnd])          EndIf       EndFor    EndFor    ListenedKeys.REMOVE(broadkeys)    For
i←ListenedKeys         CanBdcastNodes.APPEND(KeyToId[i])    EndFor    CleanNodes←nodes.REMOVE(CanBdcastNodes)    Return
CanBdcastNodes,CleanNodes

Next, we need to classify *PreHon&Mal* further. Considering the malicious vehicles and Sybil vehicles have a success rate of 0.6, while the probability of communication between normal vehicles and Sybil vehicles is 0, we further classify *PreHon&Mal* through Sybil vehicles.

The environment easily influences vehicle communication. Despite using the dichotomy method in Section 4.1 to ensure effective communication, a few normal or malicious vehicles may fail to connect with *Tru-Nodes* in each round. These vehicles might be wrongly categorized in *PreSybs1*. To reduce the impact of these vehicles on subsequent detection, we use the parameter $\gamma_{syb}$ = 0.1 to select the vehicle with a lower Pval value in *PreSybs1* and record them as *PreSybs1*’. We use *PreSybs1*’ to assist in the classification of *PreHon&Mal*.

We first assign the communication status to *PreSybs1*’ through the dichotomy method in Section 4.1 and broadcast status to *PreHon&Mal*. Then, after sufficient communication between *PreSybs1*’ and *PreHon&Mal*, the RSU extracts the listened lists of *PreSybs1*’. Because there are Sybil vehicles in *PreSybs1*’ in the broadcast state, this means that they have *Keys*. Therefore, we need to remove the *keys* in this section from the listened lists and use the remaining *keys* to search for the corresponding vehicles, categorizing the listened vehicles as *PreMals1*, and the remaining vehicles as *PreHons1*, which achieves fine-grained classifications of “original Sybs”.

#### 4.4.2. Subcategory “Original Hons”

We analyze that misclassified vehicles in “original Hons” may be malicious or Sybil vehicles in Section 4.1. Their characteristic is that there is a possibility of successful communication with Sybil vehicles, while the probability of successful communication between normal vehicles and Sybil vehicles is 0. Therefore, our strategy is to first screen out the vehicles in “original Hons” that can interact with Sybil vehicles and set them as predicted Sybil vehicles and malicious vehicle group (*PreSyb&Mal*), and the remaining vehicles are classified as predicted normal vehicle group2 (*PreHons2*). Secondly, *PreSyb&Mal* is subdivided into predicted malicious vehicle group2 (*PreMals2*) and Sybil vehicle group2 (*PreSybs2*).

Because Sybil vehicles can communicate with malicious or Sybil vehicles but cannot communicate with normal vehicles, we use *PreSybs1*’ in Section 4.1 to assist in classification. We still allocate the status of *PreSybs1*’ through the dichotomy method and set “original Hons” to the broadcast status. After sufficient communication between the “original Hons” and *PreSybs1*’, we collect and analyze the *keys* listened by *PreSybs1*’. We remove the *keys* assigned to *PreSybs1*’ from the listened lists and use the remaining *keys* to search for vehicles. These vehicles are classified as *PreSyb&Mal*, and the remaining vehicles are classified as *PreHons2*. The process of screening out *PreSyb&Mal* by *PreSybs1*’ is shown in Algorithm 2:
**Algorithm 2** Screening vehicles that can be listened to.detectCanBeLisdNodes(nodes,PreCleanSybs)    CanBeLisdNodes←[]    NodeSum←nodes+PreCleanSybs    $Rounds\leftarrow 2*\log_{2}\left(len\left(NodeSum\right)\right)$    For
i←Rounds       For
j←PreCleanSybs          If
CommPlan[j,rnd]≠ ”listen”              broadkeys.APPEND(ConnSim[j,rnd])          Else              ListenedKeys.APPEND(ConnSim[j,rnd])          EndIf       EndFor    EndFor    ListenedKeys.REMOVE(broadkeys)    For
i←ListenedKeys         CanBeLisdNodes.APPEND(KeyToId[i])    EndFor    CleanNodes←nodes.REMOVE(CanBeLisdNodes)    Return
CanBeLisdNodes,CleanNodes

Next, we will further classify *PreSyb&Mal*. Considering the possibility of successful communication between malicious vehicles and normal vehicles, while the probability of communication between Sybil vehicles and normal vehicles is 0, our goal is to extract vehicles as close as possible to Hon to assist in verification. Therefore, after updating the Pval value of *PreHons2*, we extract the vehicles with the top 0.1 Pval values and mark them as *Tru-Nodes*” (based on the experience of extracting *Tru-Nodes* in Section 4.4.1). And set we *Tru-Nodes*” to the listening state and *PreSyb&Mal* to the broadcasting state. After sufficient communication between these two groups, we extract the listening list of *Tru-Nodes*”, and classify the monitored vehicles as *PreMals2* and the remaining vehicles as *PreSybs2*, thereby completing the fine-grained classification of “Original Hons”.

According to Section 4.4.1 and Section 4.4.2, our detection results for Sybil vehicles are composed of *PreSybs1* and *PreSybs2*, the detection results for malicious vehicles are composed of *PreMals1* and *PreMals2*, and the detection results for normal vehicles are composed of *PreHons1* and *PreHons2*. Our goal is to trace malicious vehicles and improve the precision of Sybil vehicles.

#### 4.4.3. Update of Pval Value

In order to reduce the impact of Sybil vehicles or malicious vehicles on the Pval of normal vehicles, when there is a suspicious vehicle, we ignore the edge formed by the suspicious vehicle and use Equation (Equation 4) to recalculate the vehicle’s credibility. In this way, reducing the impact of suspicious vehicles on the Pval of normal vehicles can also make the Pval of suspicious vehicles increasingly extreme. The algorithm for updating Pval is shown in Algorithm 3.
**Algorithm 3** Updating the Pvals.ReCalcuPval(Suspinodes,nodes)    IdtoPval←[]    If
nodes∉Suspinodes        idtoPval←nodePval(idtoEdges(nodes),Suspinodes)    EndIf    Return
idtoPval

## 5. Experiments

We conduct simulation experiments to verify the effectiveness of our proposed scheme. We evaluate the proposed scheme under different Sybil attack densities, malicious vehicle densities, and vehicle density attack scenarios.

In designing our detection mechanism, we realize the importance of scalability to ensure the long-term effectiveness of the system and to adapt to future changes. Sybil Vehicle Detection and Malicious Vehicle Traceability adopt a modular design ideology, allowing each functional module to operate independently and flexibly expand and combine to meet the needs of CAV systems of varying sizes and complexities. Regarding Sybil vehicle detection, when a vehicle executes the broadcast of a beacon packet, the module analyzes the key list reported by the vehicle to identify potential Sybil vehicles accurately. Regarding tracing malicious vehicles, the detected Sybil vehicles are assigned states and traced based on their collusive behavior. Our scheme also has good openness. If a better Sybil vehicle detection algorithm appears, we can easily replace the existing detection module without making large-scale changes to the system. The information table of our experimental equipment is shown in Table 2.

### 5.1. Simulation Design

Sybil attacks have a greater impact on areas with high traffic densities, such as being more likely to cause vehicle collisions. Therefore, we analyze urban areas with higher traffic densities. Veins [43] is an open-source framework for vehicle network simulation. We use the Veins simulator to simulate the running state of the car, which is based on the road traffic simulator SUMO and the event network simulator OMNeT++. As shown in Figure 9, we select a route map for some areas of Haidian Island through OpenStreetMap and use SUMO traffic scenarios to describe vehicle trajectories. Our simulation parameters are shown in Table 3.

We randomly select a circle with a center position and radius *R* as our detection range. Vehicles in the range are to be detected. According to the Distances and Communication Success Probabilities model in Section 4.3.2, to ensure that the vehicle is within a high probability of communication success, we set *R* to 200 m. This allows vehicles to prove their physical presence by broadcasting. When a Sybil vehicle is instructed to broadcast, a malicious vehicle in a listening state replaces the Sybil vehicle for broadcasting. When a Sybil vehicle is instructed to listen, the malicious vehicle in an actual listening state shares the listened *key* with the Sybil vehicle.

### 5.2. Evaluation Metrics

We evaluate the performance of our detection mechanism using three indicators: precision, recall, and F1-score (F1). Precision is the ratio of correctly predicted positive samples to the total number of predicted positive samples, recall is the ratio of predicted positive samples among actually positive samples, and the F1-score is the harmonic mean of precision and recall rates. The precision and recall calculation formulas for Sybil vehicles are represented by Equations (Equation 5) and (Equation 6), respectively:(5)$$\begin{array}{ccc} P_{s} & = & \frac{TP_{s}}{TP_{s}+FP_{s}} \end{array}$$
(6)$$\begin{array}{ccc} R_{s} & = & \frac{TP_{s}}{TP_{s}+FN_{s}} \end{array}$$

The precision and recall calculation formulas for malicious vehicles are expressed as Equations (Equation 7) and (Equation 8), respectively:(7)$$\begin{array}{ccc} P_{m} & = & \frac{TP_{m}}{TP_{m}+FP_{m}} \end{array}$$
(8)$$\begin{array}{ccc} R_{m} & = & \frac{TP_{m}}{TP_{m}+FN_{m}} \end{array}$$

The F1 calculation formulas for Sybil vehicles and malicious vehicles are represented by Equations (Equation 9) and (Equation 10), respectively:(9)$$\begin{array}{ccc} F1_{s} & = & \frac{2\times P_{s}\times R_{s}}{P_{s}+R_{s}} \end{array}$$
(10)$$\begin{array}{ccc} F1_{m} & = & \frac{2\times P_{m}\times R_{m}}{P_{m}+R_{m}} \end{array}$$

The overall precision calculation formula for Sybil vehicles and malicious vehicles is as follows (Equation 11):(11)$$\begin{array}{ccc} P_{s\&m} & = & \frac{TP_{s}+TP_{m}}{TP_{s}+FP_{s}+TP_{m}+FP_{m}} \end{array}$$

The overall recall calculation formula for Sybil vehicles and malicious vehicles is Equation (Equation 12):(12)$$\begin{array}{ccc} R_{s\&m} & = & \frac{TP_{s}+TP_{m}}{TP_{s}+FN_{s}+TP_{m\;}+FN_{m}} \end{array}$$

The overall F1 calculation formula for Sybil vehicles and malicious vehicles is Equation (Equation 13): (13)$$\begin{array}{ccc} F1_{s\&m} & = & \frac{2\times P_{s\&m}\times R_{s\&m}}{P_{s\&m}+R_{s\&m}} \end{array}$$

*Note*: The Hunting scheme [17] does not classify malicious vehicles. We select malicious vehicles from “original Sybs” as the predicted malicious vehicles in the Hunting scheme.

## 6. Performance Evaluation

In order to reduce the situation where normal vehicles cannot generally run due to detection errors, our goal is to increase the recall as much as possible while ensuring precision. Our comparative schemes are the Hunting scheme [17] and the Eliminate scheme [41]. The Hunting scheme [17] is chosen due to its graph-based detection mechanism. This mechanism avoids the need for extensive data training like machine learning-based schemes. It offers efficient detection and a high detection rate for Sybil vehicles. Our work is based on the Hunting scheme [17] to trace malicious vehicles and has improved the detection rate of Sybil vehicles compared to this scheme. On the other hand, the Eliminate scheme [41] is chosen because although both this scheme and our proposed traceability mechanism are based on beacon packet analysis, this scheme utilizes the unique signal source of beacon packets to trace malicious vehicles. In contrast, our traceability mechanism traces malicious vehicles through the possibility of successful communication between Sybil vehicles and malicious vehicles. We mainly evaluate the solution’s performance from Sybil vehicle proportions, malicious vehicle proportions, and vehicle densities.

### 6.1. Increasing the Proportion of Sybil Vehicles

We use different attack densities of Sybil vehicles to verify the effectiveness of the schemes. The normal and malicious vehicles ratio is 5:1, with 25 and 5 vehicles, respectively. The Sybil vehicle ratios are 5%, 10%, 20%, 30%, and 40% for performance verifications.

As the proportion of Sybil vehicles increases, malicious vehicles cannot continuously maintain a large number of Sybil vehicles, making them more susceptible to exposure. Therefore, all three schemes can stably detect Sybil vehicles.

As shown in Figure 10, on the detection rate of Sybil vehicles, the graphs $P_{s}$, $R_{s}$, $F1_{s}$ reflect the precision rate, recall rate, and F1 value of the three schemes for Sybil vehicle detection under different Sybil vehicle attack densities, respectively. Among them, the Hunting [17] scheme has a stable overall performance in the three metrics. This is because as the number of Sybil vehicles increases, the malicious vehicles cannot maintain the gradually increasing number of Sybil vehicles, resulting in a decrease in the likelihood of them successfully establishing connections with normal vehicles. This makes the trustworthiness of Sybil vehicles behave more extremely in the neighbor graph. As a result, there is a slight upward trend in the three metrics of the scheme for different Sybil vehicle attack densities. It also reveals that the Hunting scheme can maintain a relatively stable performance when dealing with different densities of Sybil vehicles. The point strategy of the Eliminate scheme [41] depends on the designation of neighboring vehicles. As the density of Sybil vehicles increases, the probability that neighboring vehicles will be mixed with Sybil vehicles increases. However, since a Sybil vehicle is just a beacon packet, it needs to depend on the maintenance of malicious vehicles to impact the decision of the points strategy. Therefore, this scheme performs more consistently in the three metrics for detecting Sybil vehicles. Our Trace scheme is based on the Hunting scheme for the fine-grained classification of predicted Sybil vehicles. Even when the density of Sybil vehicles is low, our scheme maintains a high detection rate compared to the other two, suggesting that it is more sensitive to stealthy attacks. For $F1_{s}$, our scheme also improves by 0.9% and 3% over the Hunting and Eliminate schemes, respectively, further proving its superiority in Sybil vehicle detection.

On the detection rate of malicious vehicles, Figures $P_{m}$, $R_{m}$, $F1_{m}$ then reflect the precision rate, recall rate, and F1 value of the three schemes for malicious vehicle detection, respectively. The Hunting [17] scheme takes into account both the selfish behavior of malicious vehicles, prioritizing their broadcasting tasks to avoid exposure, and the clustering tendency between malicious and Sybil vehicles, resulting in reduced interaction between them. However, with the increase in Sybil vehicles, malicious vehicles still need to maintain more Sybil vehicles, so the detection rate of malicious vehicles in this scheme is on the rise. Nevertheless, the selfish behavior of malicious vehicles leads to a poor overall detection rate. Furthermore, because the Hunting scheme has a good detection rate for Sybil vehicles, the $P_{s}$ in Figure 10 is close to 1. Nevertheless, the screening rate of malicious vehicles from the “original Sybs” is very low, which leads to approaching 0. We use max(x,1) to treat the denominator non-0, so the calculated $P_{m}$ approaches 0 and $P_{s\&m}$ approaches $P_{s}$.

In the Eliminate [41] scheme, which utilizes suspicious Sybil vehicles to locate the source of packets that are further sent as malicious vehicles, the detection rate of malicious vehicles remains stable, as the overall detection rate of Sybil vehicles remains stable. However, after detecting Sybil vehicles, our scheme leverages the conspiracy between Sybil vehicles and malicious vehicles to trace the malicious vehicles. This is despite the malicious vehicle randomly reducing its interactions with the Sybil vehicle while maintaining the Sybil vehicle to reduce the risk of being detected. The communication state and number of rounds we set in Section 4.4.1 and Section 4.4.2 provide ample opportunities for their interactions. For $F1_{m}$, our traceability mechanism is improved by 93.9% and 4.3% compared to the Hunting [17] and Eliminate [41] schemes, respectively. The effectiveness of using the conspiracy behavior between Sybil and malicious vehicles to trace malicious vehicles is demonstrated.

Figures $P_{s\&m}$, $R_{s\&m}$, and $F1_{s\&m}$ comprehensively present the three schemes’ precision, recall, and F1 values regarding the overall detection of Sybil vehicles and malicious vehicles. Our scheme performs well in both Sybil vehicle detection and malicious vehicle tracing. This result is the uniqueness of our use of fine-grained classification. Regarding detecting Sybil vehicles, the Trace scheme and Hunting scheme perform better than the Eliminate scheme. This is mainly due to their use of constructed graphs, which are more accurate than distance-dependent detection. The communication relationship between nodes is constructed as a graph, which enables more accurate identification of Sybil vehicles. However, distance-based detection is more susceptible to interference from various factors, such as the environment and communication delays, which can decrease precision.

### 6.2. Increasing the Proportion of Malicious Vehicles

Due to the fact that Sybil vehicles are only beacon packets, they need to be maintained by a malicious vehicle in order to exhibit the characteristics of a normal vehicle. Therefore, we want to verify whether an increase in the proportion of malicious vehicles will affect the detection rate of Sybil and malicious vehicles. Therefore, we adopt a ratio of 5:1 for normal vehicles and Sybil vehicles, with 25 and 5 vehicles, respectively, and 5%, 10%, 20%, 30%, and 40% for malicious vehicles, respectively.

As shown in Figure 11, the graphs $P_{s}$, $R_{s}$, and $F1_{s}$ reflect the precision rate, recall rate, and F1 value of the three schemes for Sybil vehicle detection under different malicious vehicle attack densities, respectively. As the proportion of malicious vehicles increases, the detection rate of the Hunting scheme on Sybil vehicles shows a significant decreasing trend. The reason is that the increase in the density of malicious vehicles enables more malicious vehicles to help Sybil vehicles broadcast, which makes Sybil vehicles maintain a high level of confidence in the neighbor graph. As a result, the Hunting scheme increases the likelihood of misclassifying this number of Sybil vehicles into normal vehicles, leading to a decrease in the detection rate of Sybil vehicles. In the Eliminate [41] scheme, as the proportion of malicious vehicles increases, the voting probability of malicious vehicles participating in the integral strategy will fluctuate, resulting in a slight fluctuation in the detection rate of Sybil vehicles in this scheme. Although our scheme has a fluctuating trend, overall, it maintains a high detection rate. That is because as the proportion of malicious vehicles increases, it increases the probability of malicious vehicles maintaining Sybil vehicles continuously, making a small number of Sybil vehicles behave less extremely. Our strategy for less extreme Sybil vehicles is to extract trusted groups with higher Pval in through Section 4.4.2 instead of directly using and then further classify *PreSyb&Mal*. The advantage of this approach is that it can reduce the impact of Sybil vehicles successfully disguised by malicious vehicles, as their Pval is difficult to exceed the Pval of normal vehicles. However, we also know the limitations, where malicious vehicles may appear in trusted groups when they do not maintain Sybil vehicles. In the future, we will consider implementing additional authentication mechanisms, such as private key authentication, on the extracted trusted groups to ensure they contain only normal vehicles. For $F1_{s}$, our traceability mechanism has improved by 2.3% and 3.47% compared to the Hunting [17] and Eliminate [41] schemes, respectively.

In terms of the detection rate of malicious vehicles, as the proportion of malicious vehicles increases, more malicious vehicles in the Hunting [17] scheme can maintain Sybil vehicles, increasing their interaction. Therefore, there is an overall upward trend. However, due to the selfish behavior of malicious vehicles, this scheme only affects the credibility of malicious vehicles through Sybil vehicles, which is far from achieving a high detection rate for malicious vehicles. For $F1_{m}$, our traceability mechanism is increased by 91.87% and 3.52% compared to the Hunting [17] and Eliminate [41] schemes, respectively. All three scenarios show a decreasing trend when the percentage of malicious vehicles increases. This is mainly due to the disadvantage of relying on detecting neighboring vehicles, which are prone to vote manipulation by malicious vehicles.

### 6.3. Increasing the Vehicles Density

Since our proposed traceability mechanism is based on vehicle location information, we would like to verify whether the detection rate of Sybil vehicles and malicious vehicles is affected when the vehicle density increases. Therefore, we adopt ratios of 5:1 between normal and malicious vehicles, with Sybil vehicles accounting for 20% of both.

As shown in Figure 12, graphs $P_{s}$, $R_{s}$, and $F1_{s}$ reflect the precision rate, recall rate, and F1 value of the three schemes for Sybil vehicle detection, respectively. Figures $P_{m}$, $R_{m}$, and $F1_{m}$ reflect the precision rate, recall rate and F1 value of the three schemes for malicious vehicle detection, respectively. And Figures $P_{s\&m}$, $R_{s\&m}$, and $F1_{s\&m}$ combine the precision, recall, and F1 values of the three schemes for Sybil vehicles and malicious vehicles in general. Among them, the Eliminate [41] scheme is most affected by vehicle density and shows a decreasing trend overall. This is mainly due to the fact that the spatio-temporal model adopted by the Eliminate scheme is easily affected by signal transmission. When the vehicle density increases, the complexity and interference of signal transmission also increase, leading to a decrease in the accuracy of the spatio-temporal model. To compensate for this shortcoming, the authors propose an integration strategy that attempts to reduce the error through the information of neighboring vehicles. However, this integration strategy is more sensitive to vehicle density and is prone to false alarms in the case of high vehicle density. When detecting Sybil vehicles, if the theoretical distance between the Sybil vehicles and the vehicles to be detected has the minimum error compared to the spatio-temporal model distance, the Sybil vehicle will be mistaken for the source of the beacon packet and classified into normal vehicles. In this case, Sybil vehicles and malicious vehicles cannot be detected. In addition, when there is a suspicious Sybil vehicle to search for the source of the beacon packet, if the theoretical distance between the normal vehicle and the vehicle to be detected has the minimum error compared to the spatio-temporal model distance, the normal vehicle will be mistaken for the source of the Sybil vehicle and classified as a malicious vehicle. Our traceability mechanism focuses on proving the physical existence between vehicles through broadcasting *keys*. The greater the density of vehicles, the lesser the possibility that successful communication between the vehicles is affected by the distance, and the more advantageous our mechanism. Our traceability mechanism is increased by 0.4% and 3.22%, respectively, compared to the Hunting [17] and Eliminate [41] schemes in $F1_{s}$, and by 95.39% and 3.29% respectively compared to the Hunting [17] and Eliminate [41] schemes in $F1_{m}$.

Due to our traceability mechanism’s focus on tracing malicious vehicles in Sybil attacks, we compare the detection rates of three schemes for malicious vehicles using a table as shown in Table 4. Under the three attack densities of Sybil vehicle proportions, malicious vehicle proportions, and vehicle densities, the arithmetic mean of $P_{m}$, $R_{m}$, and $F1_{m}$ is calculated respectively and expressed with $AvgP_{m}$, $AvgR_{m}$, and $AvgF1_{m}$.

Table 4 shows that the precision of our proposed scheme for malicious vehicles is as high as 98.53%. This excellent performance is mainly attributed to the stability of our scheme, which is not easily affected by the fluctuation of vehicle density. Meanwhile, when the number of malicious vehicles increases, we effectively reduce the risk of being disguised as Sybil vehicles by malicious vehicles by extracting trusted groups, thus ensuring efficient detection of malicious vehicles. In contrast, the Hunting scheme has a lower detection rate for malicious vehicles and is still deficient in malicious vehicle tracing. The detection rate of the Eliminate scheme is slightly lower than that of the Trace scheme because the proposed spatio-temporal model is easily affected by environmental disturbances and vehicle density.

In order to evaluate the overall performance of the three schemes, we calculate the arithmetic mean values of $P_{s\&m}$, $R_{s\&m}$, and $F1_{s\&m}$ under the proposed three attack densities. Then, we express them in $AvgP_{s\&m}$, $AvgR_{s\&m}$, and $AvgF1_{s\&m}$. A comparison of the overall performance and detection time of the three schemes is shown in Table 5.

The comprehensive analysis of the experimental results shows that our proposed traceability mechanism exhibits good classification performance results in detecting Sybil vehicles and malicious vehicles under three attack densities, and our scheme is not easily affected by vehicle density. The average precision of Sybil and malicious vehicles is as high as 98.79%, which is about 5% higher than the two comparative schemes. Our scheme also exceeds 96% in terms of overall average recall and F1, which proves that our traceability mechanism is feasible for detecting Sybil vehicles and tracing malicious vehicles. In terms of running time, our scheme takes only 1.35 s to achieve more than 95% of the F1 value, which can satisfy the demand for real-time detection. In contrast, the Eliminate scheme is time-consuming because it needs first to analyze the packets broadcasted by all neighboring vehicles, calculate a broadcaster with the smallest distance error as the source of suspicious packets, and then make a collective decision to detect Sybil vehicles through the integral strategy. This complex processing flow poses a considerable challenge to the real-time nature of Sybil attack detection. Our scheme shows obvious advantages in detection rate, accuracy, and running time and provides a practical solution for Sybil vehicle detection and malicious vehicle tracing.

### 6.4. Threshold Analysis

We conduct an in-depth experimental analysis of the introduced thresholds, aiming to find the optimal threshold setting through data validation to improve the precision of malicious vehicle tracing further. Figure 13a demonstrates the change in the extraction probability of honest vehicles for different values of $\alpha_{Tru\text{-}Nodes}$. Figure 13b reflects the change in the extraction probability of honest vehicles for different values of $\beta_{similar}$. Figure 13c depicts explicitly the effect of different values of $\gamma_{syb}$ on the extraction probability of Sybil vehicles.

#### 6.4.1. $\alpha_{Tru\text{-}Nodes}$

According to the Pval value mentioned in Section 4.3.2, the higher the Pval value, the greater the likelihood that the vehicle is normal. The $\alpha_{Tru\text{-}Nodes}$ value affects the accuracy of *Tru-Nodes*, which is crucial for the algorithm’s performance. Therefore, this section conducts an experimental analysis on the threshold of $\alpha_{Tru\text{-}Nodes}$ at different levels. We take the value of each probability at intervals of 0.1 and conduct 100 independent repeated experiments. The results are shown in Figure 13a. It can be seen that with the increase in the $\alpha_{Tru\text{-}Nodes}$ value, the probability that the vehicle is normal is lower. This is because malicious vehicles will randomly maintain Sybil vehicles, and the Pval value of such malicious vehicles will be affected by Sybil vehicles, resulting in a lower ranking. However, when $\alpha_{Tru\text{-}Nodes}$ is less than 0.1, an effective number of vehicles will not be selected. To ensure the stability of the algorithm, the $\alpha_{Tru\text{-}Nodes}$ value starts from 0.1. When $\alpha_{Tru\text{-}Nodes}$ value = 0.1, the probability of Hon extraction is the best. So, our $\alpha_{Tru\text{-}Nodes}$ value is set to 0.1.

#### 6.4.2. $\beta_{similar}$

In the algorithm, a parameter $\beta_{similar}$ is designed to ensure that the extracted Tru-Nodes are all normal vehicles as much as possible, and the remaining vehicles after elimination by $\beta_{similar}$ constitute the Supls_Toppval_. In order to evaluate the effect of leaving as many normal vehicles as possible in the Supls_Toppval_ as well as the malicious vehicles culling effect, ProA and ProB are introduced, respectively. The formulas are as in (Equation 14): (14)$$\begin{array}{c} \left\{\begin{array}{c} ProA=\frac{{Supls}_{\mathrm{Toppval}}\cap {Tru}_{\mathrm{Hons}}}{Tru\text{-}Nodes}\\ ProB=\frac{{Supls}_{\mathrm{Toppval}}\cap {Tru}_{\mathrm{Hons}}}{{Supls}_{\mathrm{Toppval}}} \end{array}\right. \end{array}$$
where Tru_Hons_ are actual hon vehicles. Ultimately, the credibility of *Tru-Nodes* is as in Equation (Equation 15):(15)$$Cred_{Tru\text{-}Nodes}=\frac{ProA+ProB}{2}$$

As shown in Figure 13b, the higher the $\beta_{similar}$, the higher the confidence of *Tru-Nodes*. This is because by eliminating suspicious malicious vehicles in *Tru-Nodes*, the probability that malicious vehicles can maintain the normal broadcast of Sybil vehicles is lower. Thus, the probability that normal vehicles can listen to Sybil vehicles is negligible. That is, with the lower probability that normal vehicles are affected by malicious vehicles, an upward trend is shown. Hence, our $\beta_{similar}$ is set to 0.9.

#### 6.4.3. $\gamma_{syb}$

According to the Pval value mentioned in Section 4.3.2, it can be seen that the smaller the Pval value, the higher the probability that the vehicle is a Sybil vehicle, and the $\gamma_{syb}$ value designed in the algorithm affects the accuracy of *PreSybs1*’. We choose to take values for each probability at intervals of 0.1 and perform 100 independent repetitions of the experiment. We want to extract vehicles more likely to be Sybil vehicles, so the $\gamma_{syb}$ value is set to 0.1. However, a slight decrease in the $\gamma_{syb}$ value after 0.8 can be seen from Figure 13c, which means that the parameter is not very sensitive to the algorithm. Consequently, this parameter’s influence on the algorithm is not critical.

## 7. Conclusions and Future Work

Existing detection mechanisms only detect Sybil vehicles and cannot trace malicious vehicles. We propose a scheme to trace malicious vehicles based on vehicle broadcast beacon packets by analyzing the differences between Sybil vehicles, malicious vehicles, and normal vehicles. The experimental results show that under three attack densities, namely, increasing the Sybil proportion, malicious vehicle proportion, and vehicle density, the traceability mechanism achieves an average checking accuracy and completeness of 98.53% and 95.93%, respectively. Our traceability mechanism performs better than the latest solution for tracking malicious vehicles, particularly showing a more stable detection rate under high vehicle density. This implies enhanced defense against Sybil attacks from initiators, which is crucial for improving the security and reliability of intelligent transportation systems.

The malicious vehicle-tracing mechanism proposed in this paper demonstrates superior accuracy and stability in experimental comparisons with other schemes. However, it also exhibits limitations and shortcomings.
•During detection, we discovered that neighbor-based collective witnessing reduces errors from a few vehicles but is vulnerable to manipulation by malicious vehicles. Future research will focus on a detection scheme independent of neighboring vehicles, necessitating significantly enhancing individual vehicles’ detection capabilities. This implies stronger data processing and analysis for each vehicle to independently assess its surroundings, posing algorithm design and performance challenges. It necessitates deeper research and meticulous debugging to address diverse scenarios and attacks.•The proposed mechanism depends on trusted RSUs for key management in vehicle broadcasting, making it vulnerable to single-point failure. Once the RSU is attacked or malfunctions, it is easy to cause the entire detection mechanism to malfunction. Decentralization is a promising approach to alleviating this problem. It distributes key distribution and management tasks across multiple nodes, minimizing the impact of a single point of failure.•Current research focuses on simulations and lacks real-world validation. This hinders the mechanism’s feasibility and reliability in practical settings. Future studies should incorporate real data and scenario testing to ensure the mechanism’s effectiveness in real environments.

## Figures and Tables

**Figure 1 sensors-24-02153-f001:**
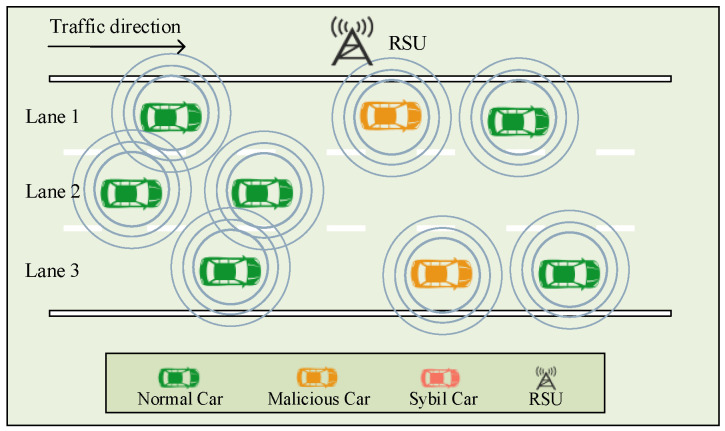
Before the Sybil attacks are launched.

**Figure 2 sensors-24-02153-f002:**
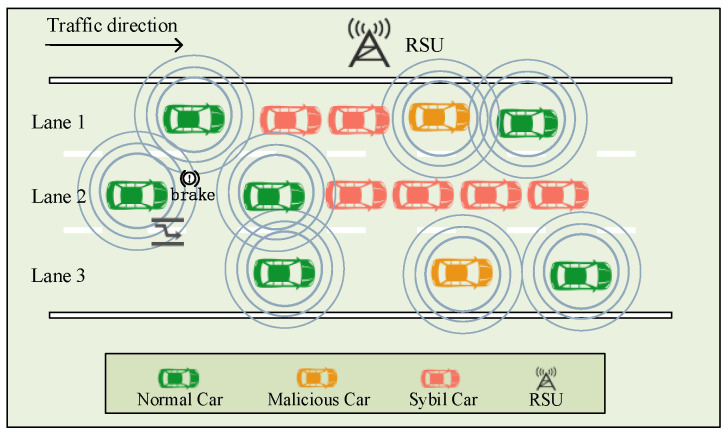
After the Sybil attacks are launched.

**Figure 3 sensors-24-02153-f003:**
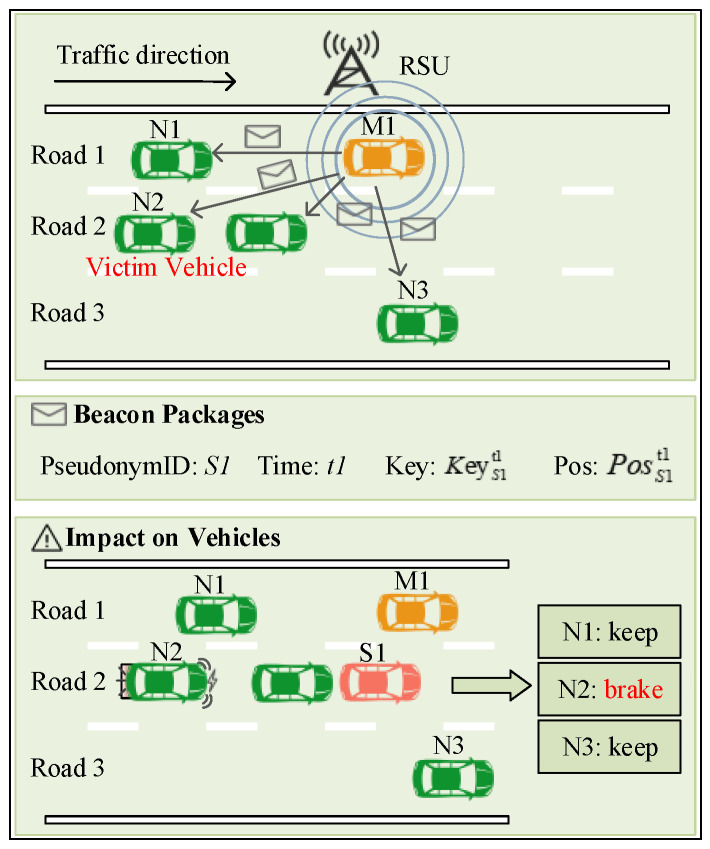
The principles and impactions of Sybil attacks.

**Figure 4 sensors-24-02153-f004:**
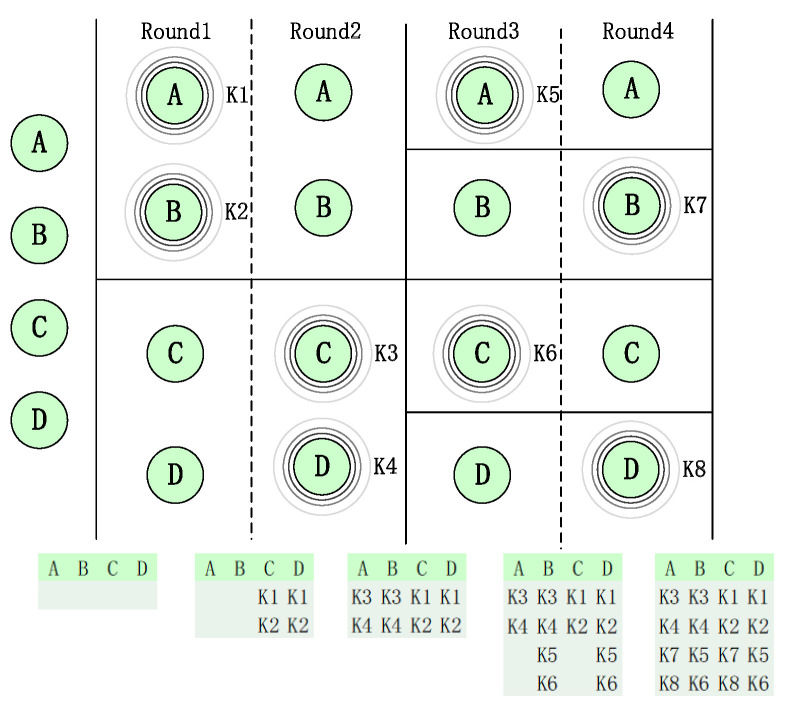
Broadcasting or listening state allocation diagram (each round is executed for *T* time; *T* is the minimum time interval).

**Figure 5 sensors-24-02153-f005:**
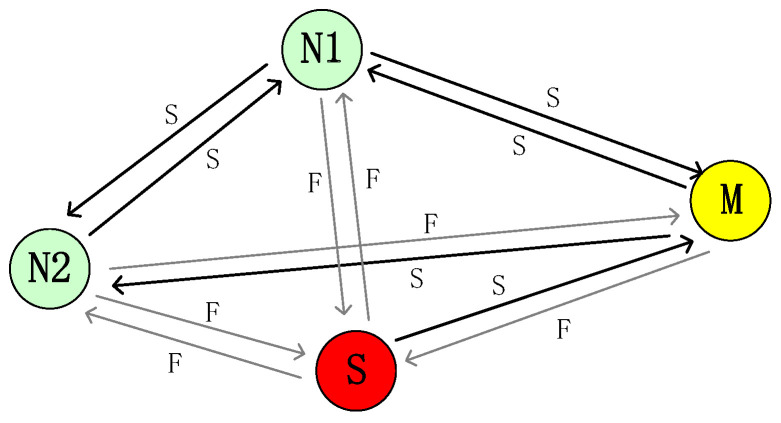
Neighbor graph.

**Figure 6 sensors-24-02153-f006:**
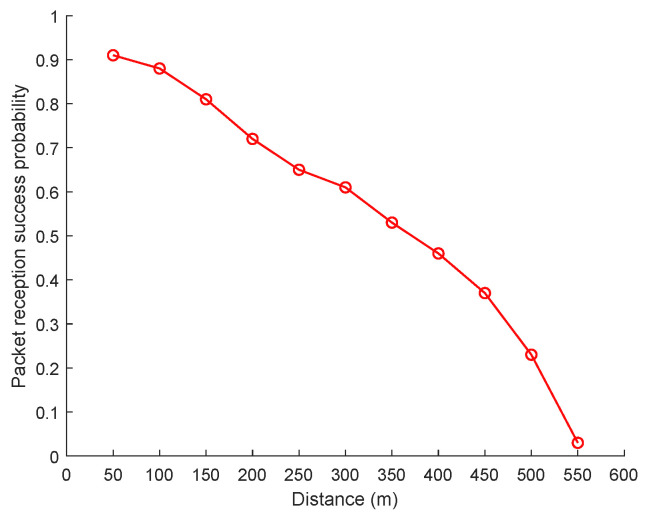
Distances and Communication Success Probabilities model.

**Figure 7 sensors-24-02153-f007:**
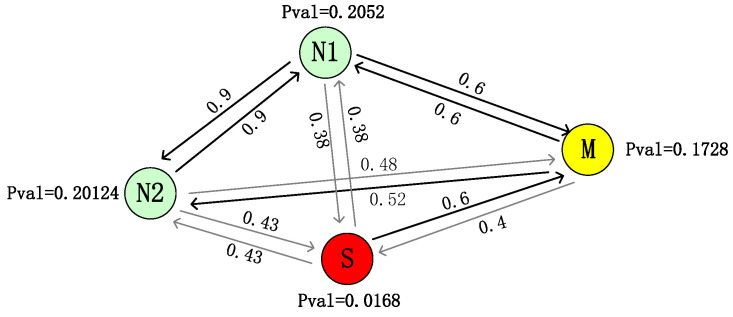
Probabilistic neighbor graph (Pval is the credibility of the vehicle).

**Figure 8 sensors-24-02153-f008:**
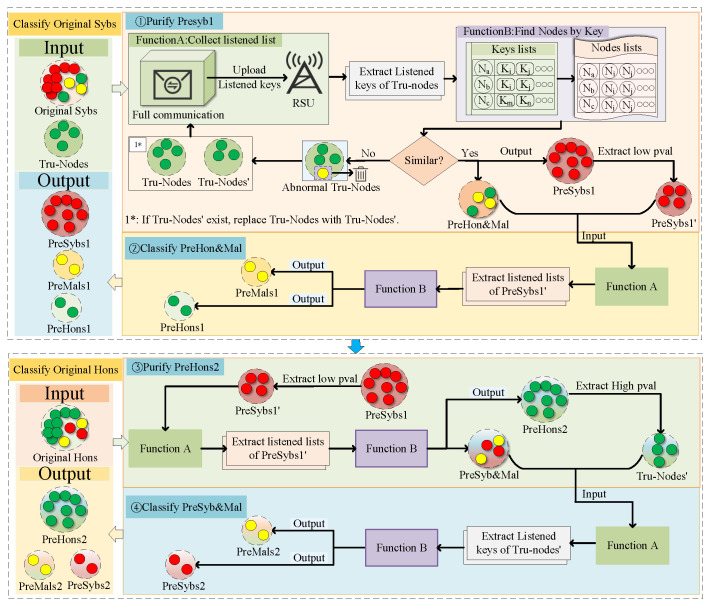
Malicious vehicle traceability mechanism diagram.

**Figure 9 sensors-24-02153-f009:**
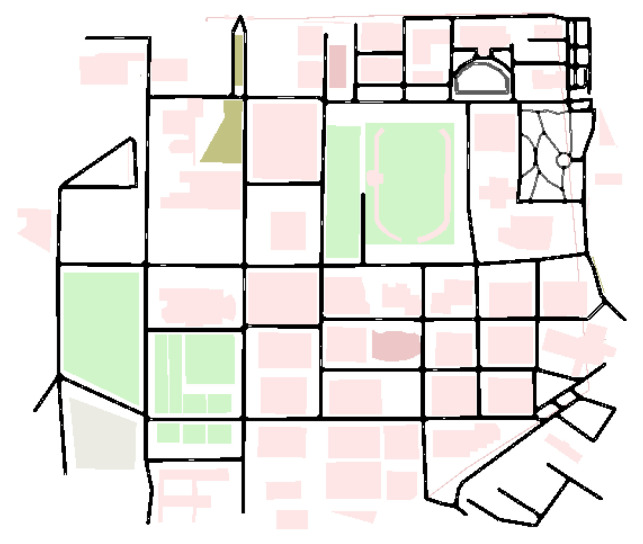
Haidian Island area.

**Figure 10 sensors-24-02153-f010:**
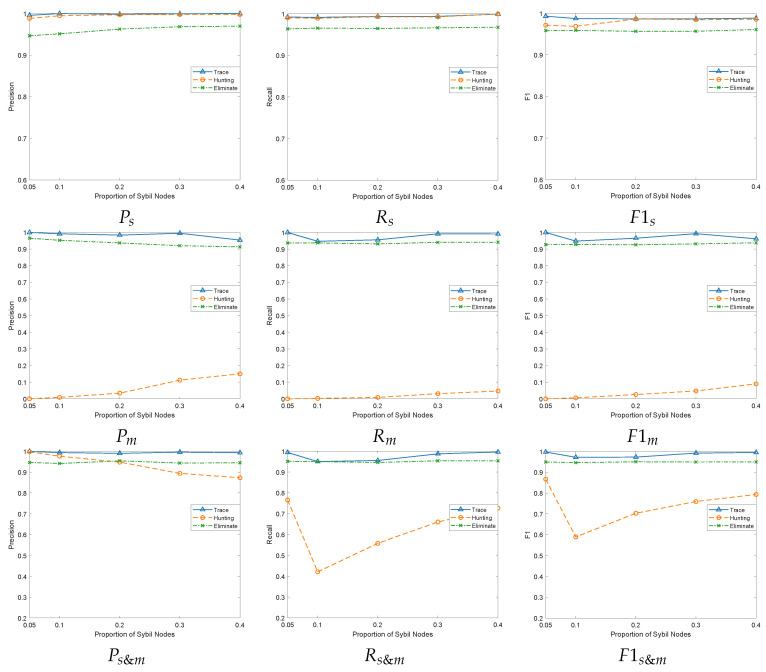
Performance comparisons under different Sybil attack densities.

**Figure 11 sensors-24-02153-f011:**
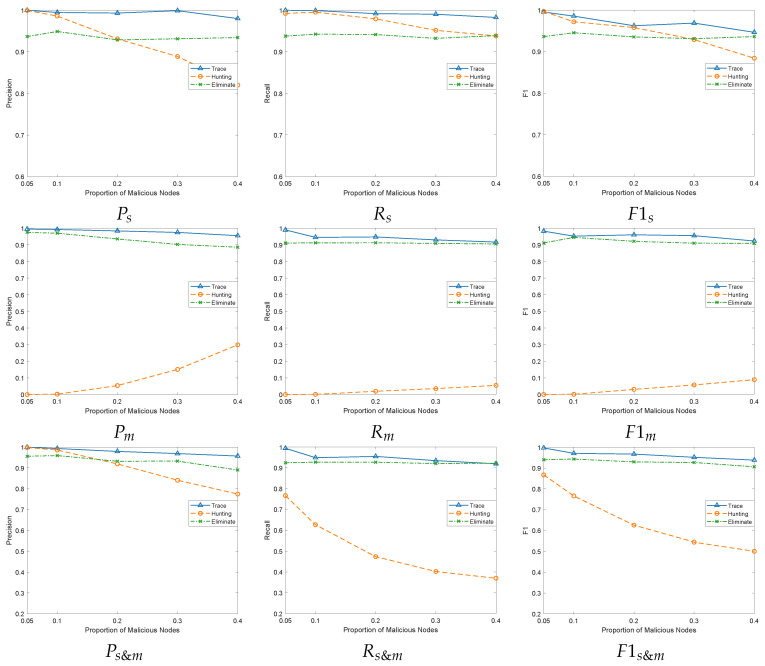
Performance comparisons under different attack densities of malicious vehicles.

**Figure 12 sensors-24-02153-f012:**
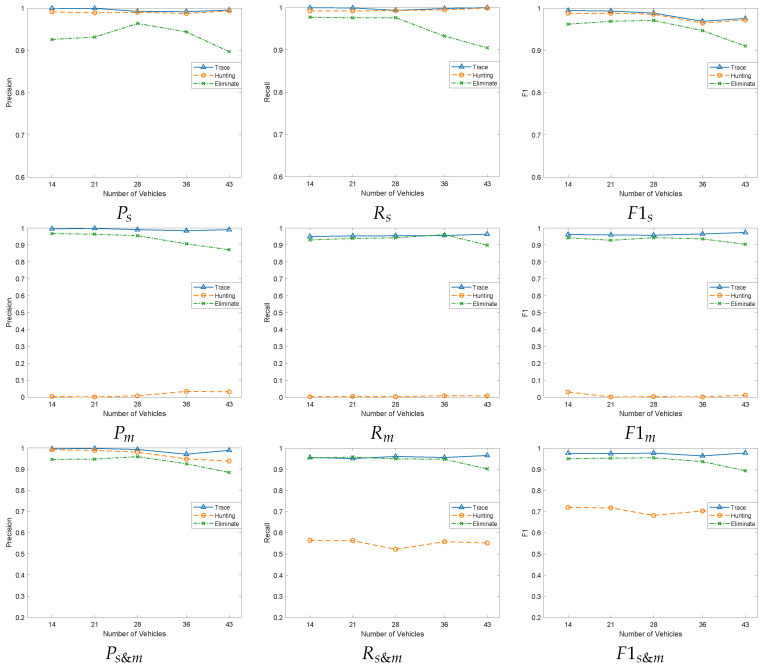
Performance comparisons under different vehicle densities.

**Figure 13 sensors-24-02153-f013:**
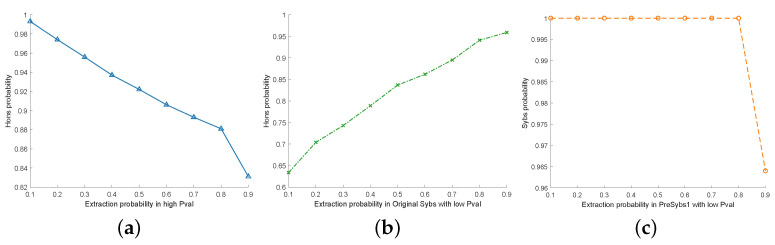
Threshold analysis. (**a**) $\alpha_{Tru\text{-}Nodes}$. (**b**) $\beta_{similar}$. (**c**) $\gamma_{syb}$.

**Table 1 sensors-24-02153-t001:** Vehicle property table.

	Physical Location	Legal Identity	Broadcast and Listen
Normal Vehicles	✓	✓	✓
Malicious Vehicles	✓	✓	✓
Sybil Vehicles	×	✓	×

**Table 2 sensors-24-02153-t002:** Experimental equipment information.

Device	Detail
CPU	Intel^®^ Core™ i7-10750H CPU @ 2.60GHz
OS	Debian GNU/Linux 11 (bullseye)
OS Type	64-bit
Memory	3.8 GB

**Table 3 sensors-24-02153-t003:** Simulation parameter settings.

Parameter	Value
Simulation Time	100 s
Attack Probability	5%, 10%, 20%, 30%, 40%
Simulation Area	400 m × 400 m
Obstacle Shadowing	Simple Path Loss Model
MAC Implementation	IEEE 802.11p
Minimum Receive Power	−110 dBm
Carrier Frequency	5.9 GHz
Noise Floor	−98 dBm
Antenna Height	1.895 m
Simulator	Veins
Path loss index	2
Obstacle loss index	0.4

**Table 4 sensors-24-02153-t004:** Comparisons of three schemes for detecting malicious vehicles.

	${\mathrm{AvgP}}_{m}$	${\mathrm{AvgR}}_{m}$	$\mathrm{AvgF}1_{m}$
Trace (ours)	**0.9853**	**0.9593**	**0.9638**
Hunting [17]	0.0591	0.015	0.3637
Eliminate [41]	0.9345	0.927	0.9264

**Table 5 sensors-24-02153-t005:** Overall performance and detection time comparison of three schemes.

	${\mathrm{AvgP}}_{s\&m}$	${\mathrm{AvgR}}_{s\&m}$	$\mathrm{AvgF}1_{s\&m}$	Time
Trace (ours)	**0.9879**	**0.9601**	**0.9725**	1.35 s
Hunting [17]	0.9371	0.5685	0.2578	0.86 s
Eliminate [41]	0.9376	0.9402	0.9385	34.59 s

## Data Availability

Data are contained within the article.

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
