# Peer review of "Sybil Attacks Detection and Traceability Mechanism Based on Beacon Packets in Connected Automobile Vehicles"

_sensors, 2024, doi:10.3390/s24072153_

Round 1

Reviewer 1 Report

Comments and Suggestions for Authors

- The authors focused on tracking the attack source of malicious vehicles by using a novel detection mechanism that relies on vehicle broadcast beacon packets. They utilize the roadside unit (RSU) that randomly instructs any vehicle pair. The vehicle’s credibility is determined by calculating the edge success probability of vehicles in the neighbor graph.

- The existing work on defending against Sybil attacks has almost exclusively focused on detecting Sybil vehicles, ignoring the traceability of malicious vehicles. As a result, they cannot fundamentally alleviate Sybil attacks.

- A beacon packet-based detection method to trace malicious vehicles based on trusted RSUs combined with information interactions performed between CAVs.

- The paper is nicely written and well organized. Few things that need to be addressed are as follows.

Long sentences produce big confusion, break them into multiple, especially in abstract and conclusion.

In introduction, the main contributions in bullet list should be short and clear. Take the rest of the text to explain your work before the bullet list.

Write and introductory sentence for '3. THREAT MODEL AND ATTACK CATEGORIES', and then go to 3.1. Same for '4.3. Construction Of Probabilistic Neighborhood Graph', for '5. EXPERIMENTS', and for '6.4. Threshold Analysis'.

- The conclusion is written well with the findings and comparative results. Future work need extension with some more key points.

- The reference list is appropriate.

- Figure 10, 11, 12, and 13, the graphs are hard to read.

Comments on the Quality of English Language

English is fine but need to check for remaining typos to be removed.

Author Response

Dear reviewers.
Thank you very much for your valuable comments and suggestions on our paper. We take your feedback very seriously and have made careful revisions based on your review comments. The detailed revisions and our responses to each comment have been organized in the attached file.
If you have any questions or need further discussion, please contact us. We look forward to continuing to receive your guidance and assistance.

Reviewer 2 Report

Comments and Suggestions for Authors

Recommended for publication in the current form

Author Response

(The authors gave the same response as above.)

Reviewer 3 Report

Comments and Suggestions for Authors

Overall write-up and Performance analysis we impressively written.

The author must write one paragraph and explain 

1- How you justify its a novel idea

". In this work, we focus on tracking the attack source of malicious  vehicles by using a novel detection mechanism that relies on vehicle broadcast beacon packets."

Previously most of the authors have done that using RSSI values, which consequently also assist in detecting and mitigating DOS/DDOS attacks, so 

please justify using 3-5 related works that this is still a novel domain to research and contribute.

Author Response

(The authors gave the same response as above.)

Reviewer 4 Report

Comments and Suggestions for Authors

The article "Sybil Attacks Detection and Traceability Mechanism based on Beacon Packets in Connected Automobile Vehicles (CAVs)" focuses on using beacon packets to detect and trace Sybil attacks in connected vehicles. The authors propose an innovative method that allows for detecting malicious vehicles and tracking their source of attack. Experimental results show high precision and recall rates, demonstrating the effectiveness of the proposed method.

While the research introduces an innovative traceability mechanism using beacon packets, there are several critical points to consider:

1. The introduction provides a general overview of the problem of Sybil attacks in CAVs and the importance of addressing these security threats. It introduces the concept of cooperative driving and the potential risks posed by malicious vehicles in the context of intelligent transportation systems. However, this introduction could benefit from citing more references to support the background information provided. Including a broader range of references from reputable sources would enhance the credibility of the introduction and demonstrate a thorough understanding of the existing research landscape on Sybil attacks in CAVs.

2. Although the precision and recall rates are highlighted as excellent, the article falls short in providing a comprehensive discussion of these results. A deeper analysis of the implications of the findings, comparison with existing methods, and potential limitations would enhance the research's credibility.

3. The article could benefit from discussing the practical implications of implementing the proposed traceability mechanism in real-world CAV systems. Addressing issues such as scalability, computational overhead, and integration with existing infrastructure would provide valuable insights for industry practitioners.

4. While the paper briefly mentions the need for future research on detection schemes independent of neighbor vehicles, it misses the opportunity to delve into the challenges, feasibility, and potential impact of such advancements. A more critical exploration of future research directions would add depth to the study.

Comments on the Quality of English Language

The English in the article appears to be well-written and coherent based on the excerpts provided. The language used is technical and specific to the field of connected vehicle security, indicating a good level of proficiency in conveying complex concepts and research findings. The sentences are structured logically, and the terminology is appropriate for the subject matter.

Author Response

(The authors gave the same response as above.)
